# Museum-Authorization of Digital Rights: A Sustainable and Traceable Cultural Relics Exhibition Mechanism

**Yun-Ciao Wang** [1], **Chin-Ling Chen** [2,3,4,*] and **Yong-Yuan Deng** [2,*]

1 National Museum of Marine Biology and Aquarium, Pingtung 94450, Taiwan; yunciao@gmail.com
2 Department of Computer Science and Information Engineering, Chaoyang University of Technology, Taichung 41349, Taiwan
3 School of Information Engineering, Changchun Sci-Tech University, Changchun 130600, China
4 School of Computer and Information Engineering, Xiamen University of Technology, Xiamen 361005, China
* Correspondence: clc@mail.cyut.edu.tw (C.-L.C.); allen.nubi@gmail.com (Y.-Y.D.)

**Abstract:** The digital rights management of museums is a mechanism that protects digital content from being abused by controlling and managing its usage rights. Traditional museums attach importance to the collection, display, research, and education functions of "objects". In response to natural or man-made disasters, people are often caught off guard, destroying material, intangible assets, and spiritual symbolism. Therefore, with the advancement of digital technology, this research is based on the mechanism of blockchain, through the authorization of cryptographic proxy re-encryption, and proposes a new method for the preservation and authorization of digital content in museums, which can effectively display, store, and promote "important cultural relics and digital archives". In this research, the Elliptic Curve Digital Signature Algorithm (ECDSA), blockchain, and smart contracts are used to design a sustainable and traceable cultural relic exhibition mechanism. The proposed scheme achieves publicly verifiable, transparency, unforgeability, traceability, non-repudiation, standardization of stored data, timeliness, etc., goals. It is the museum's preservation and innovation approach for the unpredictable future. Through appropriate preservation and management mechanisms, it has extremely important practical significance for the protection of museum collections, the inheritance of historical and cultural heritage, and the expansion of social education.

**Keywords:** museum; digitization; blockchain; digital rights management (DRM); smart contract; sustainability

## 1. Introduction

The museum has collections, displays, research, education, and dissemination of human history and cultural heritage. By collecting all kinds of material and intangible cultural heritage, museums are the biggest driving force for their development and progress, and also the embodiment of museum creativity and vitality. With the progress of society, people's material living standards are improved at the same time; cultural consumption is widely valued by people.

Take the fire of Notre Dame Cathedral in Paris on 15 April 2019, as an example, which is a form of disaster. Although there were no casualties, what cannot be replaced is that the files and data that we had been working hard on for a long time were affected by the fire, and the spirit and symbolic meaning were also destroyed. The same situation happened in Taiwan's Cloud Gate Dance Company, an outstanding modern dance performance group in Taiwan. Founded in 1973, it was the first contemporary dance company in Asia and a world-class modern dance company, enjoying a high reputation in the world [1].

In the early hours of 11 February 2008, a fire broke out in the rehearsal venue of Bali Township, Taipei County, Taiwan (now Bali District, New Taipei City). The performance props and historical data were almost destroyed. As an institution for collecting, researching, and displaying natural and human cultural heritage, museums have attracted ever

more attention. Museums, with the mission of social responsibility, collect the current social memorabilia, especially the effective management and utilization of "important cultural relics and archives information", which is the foundation of museum's construction and development, and, also, the basic task of museums.

However, due to various reasons, most people may not be able to visit their favorite museums one by one due to time and space constraints. For example, since the outbreak of the novel coronavirus (COVID-19) in early 2020, the American Alliance of Museums (AAM) recently stated that nearly one-third of the museums in the United States may be permanently closed, pointing out that museum operations are facing "extreme financial distress" [2]. Digital museums based on the development of the Internet and multimedia technology can now satisfy people's desires, enabling people to view museums around the world without leaving their homes.

Traditional cultural relics provide preservation conditions for digital storage, appreciation, and dissemination in the form of text, images, sound, and video. However, with the increasing degree of digitalization and informatization of museums, the security of digital cultural relics has become a key issue in the transmission, transaction, and reproduction of digital cultural relics. Although, the need for digitization in cultural heritage is well established, and also proven by the millions of already existing digital assets. However, if these digitalized cultural relics exhibitions lack a complete authorization verification mechanism, these digital materials will be arbitrarily spread or even falsified. As early as 2004, Triet and Due [3] posited the idea of using digital watermarking technology to provide a wider range of online services for ordinary users of the palace music of Vietnam (recognized as a world heritage site by UNESCO in November 2003) and standardizing the limited access service for online registered users. In 2006, Chen et al. [4] studied using personal mobile devices for digital rights management (DRM) of E-museums. Unfortunately, neither of them has a relatively detailed algorithm description. As for the issue of digital rights management, even in the past, Chen et al. [5–7] proposed algorithm research on the digital content rights protection technology based on mobile devices for enterprises. However, in previous literature, there is little focus on the digital content of the museum that is valuable and important. Therefore, this paper aims to be a pioneer in research on museum digital rights and proposes a new authorization mechanism.

The blockchain mechanism was proposed by Nakamoto [8] in 2008. Blockchain is essentially a new method of data management; an open, transparent, and decentralized database. Blockchain is comprised of point-to-point transmission, a consensus mechanism, a smart contract, an encryption algorithm, and a distributed data record list storage technology system. This system can realize the tamperproof data storage, traceable data viewing, and trusted point-to-point transmission to solve the problem of trust construction without third-party supervision. The core of this system is the realization of the trustworthy circulation of value and promotion of the transformation of the "Information Internet" to the "Value Internet", which can be used as the basic protocol of the "Value Internet".

Therefore, the concepts of decentralized rights management combining blockchain and digital watermarking technology [9], blockchain, and cloud services [10–13] were proposed respectively, but these studies only focus on conceptual proposals. Some authors [14–17] reveal the recent international applications of blockchain, smart contracts, and cryptography for the protection of digital rights that have put forward more specific and detailed communication protocols. Table 1 below presents a comparison of the existing digital rights management surveys.

**Table 1.** Comparison between existing digital rights management surveys.

| Authors | Year | Objective | Merit | Demerit | Approach | Technology Used | 1 | 2 | 3 | 4 |
|---|---|---|---|---|---|---|---|---|---|---|
| Chen [5] | 2008 | Mobile users can access digital content securely in the enterprise. | Focus on enterprise DRM issue | Some security analysis weak | Symmetrical, asymmetrical cryptosystem, signature | cryptosystem | N | Y | N | Y |
| Chen [6] | 2010 | Integrate content issuer, rights issuer, mobile network service provider in a platform for DRM. | Focus on mobile user DRM | Some security analysis weak | Symmetrical, asymmetrical cryptosystem, signature | cryptosystem | N | Y | N | Y |
| Chen et al. [7] | 2014 | Support users use a mobile device to access cloud service. | Focus on mobile user DRM | Some security analysis weak | Symmetrical, asymmetrical cryptosystem, signature | cryptosystem | N | Y | N | Y |
| Ma et al. [14] | 2018 | Blockchain-based digital rights management. | Detail implemented DRM in a blockchain platform | Transparency Data format standardization, Timeless not applicable | AES, ECC, ECDSA | Ethereum, Smart Contract | Y | N | N | Y |
| Ma et al. [15] | 2018 | A master-slave blockchain DRM for pervasive computing. | Detail implemented DRM in a blockchain platform | Security analysis weak | PoW and PoC consensus | Ethereum, Smart Contract | Y | Y | N | N |
| Ma et al. [16] | 2020 | Blockchain-based decentralized trust management and secure usage control scheme of IoT big data is proposed. | Detail implemented DRM in a blockchain platform | No comparison with other approaches | Consensus | Ethereum, Smart Contract | Y | Y | Y | N |
| Hassan et al. [17] | 2020 | A protecting video and audio DRM is proposed. | Low cost, availability, platform independently, reliability and scalability | Security analysis weak | AES, ECC algorithms | Cryptography | N | Y | Y | N |

1: Blockchain-focused, 2: Architecture/Framework given, 3: Categorization of approaches, 4: Security analysis, Y: Yes, N: No. (DRM: digital rights management, ECDSA: Elliptic Curve Digital Signature Algorithm, AES: Advanced Encryption Standard, ECC: Elliptic Curve Cryptography, PoW: Proof of Work, PoC: Proof of Capacity, IoT: Internet of Thing).

Although museums have many advantages in the use of digitalization, they will also encounter some problems, especially for important cultural relics and archive information. The protection of intellectual property rights is one of them. As digital museums put exhibits on the Internet for browsing, anyone can download, modify, and disseminate them. As a cultural heritage carrier, museum archives have certain privacy requirements.

However, the advent of the digital era has exacerbated the excessive sharing of museum archives information, resulting in frequent leakage of such information. The source channels of museum archives information are complex, and incomplete information collection occurs from time to time. If there is no consensus mechanism, large and complex information may be fragmented, and as authentication is difficult, it could result in unclear property rights. Besides, there are other problems, such as the cumbersome authentication methods of existing museum archives information, which lead to the user's poor experience and inhibit the effective use of museum social functions. Now, when exhibits are digitized, the existing mature digital watermarking technology and blockchain technology can solve this problem. Museums use this technology to label their digital exhibits with their copyright information and secure them through an authorization mechanism.

The traceability and tamperproof characteristics of blockchain technology can not only realize the effective "tracking" of the infringement of museum archive information due to excessive sharing but also prevent the privacy information of museum archives based on the encryption function. The distributed ledger technology, decentralization, autonomy, and other characteristics of blockchain technology can solve the complicated process and identification difficulties in the traditional museum archive information screening and authentication process, forming a kind of information based on an "address" instead of "personal identity" consensus mechanism. By setting up the "timestamp" module to build a smart authentication service program, a piece of information given by the authentication material can be recorded into the "timestamp" on the blockchain and confirm that the recorded piece of information can be open to public inquiry, thereby improving the efficiency of the "smart authentication" of museum archives. The trustlessness and smart contract features of blockchain technology can realize the synchronization and mutual verification of museum archive information, ensure that different information can automatically and programmatically flow safely, as well as easily and conveniently checking tamperproof file information. Blockchain technology can effectively solve several problems in current museum archive information management and construction.

At present, many institutions have explored the application of blockchain technology in museums. The Russian National Tretyakov Gallery uses blockchain technology to make the "world" a sponsor of the digital construction of the museum and a collector of gallery art, enabling individuals or companies to use the platform for the Tretyakov Art Gallery collection sponsorship for digitalization. Switzerland's Zug launched the world's first online trading and sponsorship platform for digital art masterpieces created by blockchain technology to make up for the shortcomings of the current traditional art market operation mode. China's Baidu Super Chain and Baidu Baike have established a "Culture, Museum and Art Chain" based on blockchain technology and promoted the online collection of 246 museums in the Baike Museum project online, jointly promoting the confirmation and maintenance of online collection copyright and exploring online collection digital copyright trading methods [18].

In short, the application of blockchain in museums is still in the experimental stage. The World Cultural Heritage Organization released a white paper on the cultural relics blockchain in 2019, stating that with the deepening of the application of blockchain technology and the continuous updating of management ideas in the Cultural, Museum, and Art industry, blockchain technology will play a revolutionary role in related fields, such as cultural relics collection, auction, evaluation, research, rescue, excavation, protection, and museum drafting and rating. This research is based on blockchain technology, focusing on the preservation, authorized display, and security of cultural relics, and constructing

a museum's design and application of digital rights management, thereby achieving the following goals:

(1) Publicly verifiable [5,6,14–16]—The authorized on-chain information of cultural relics and important files can be disclosed, and the identity of the applicant can be verified through the verification of digital certificates.

(2) Transparency [5,14–16]—The authorization information for cultural relics and important files is transparent. Based on the open transparency of information on the chain, it can obtain trust between nodes at low cost, realize the sharing of digital content resources, and achieve the mission of museum social education.

(3) Unforgeability [14–16]—The new decentralized model of blockchain combined with timestamp technology makes it impossible to arbitrarily change the data on the chain, thus effectively solving the trust problem.

(4) Traceability [5,14–16]—The source and authorization mechanism of cultural relics and important file information use timestamp and signature mechanisms to achieve traceability management.

(5) Non-repudiation [6,14]—The use of timestamp and signature mechanisms enables the implementation of the legal authorization mechanism.

(6) Standardization of stored data [5,6]—The effective classification of digital content and formatting on the chain help to effectively manage the copyright of digital rights so intellectual property rights can be protected.

(7) Timeliness [5,17]—There is a good management mechanism to realize the sharing of digital content resources. The license is timeliness and will not be monopolized.

The rest of this article is organized as follows. Section 2 presents preliminary knowledge. Section 3 deals with the proposed sustainable and traceable cultural relic display mechanism. Section 4 presents an analysis of the proposed scheme. In Section 5, the discussions and comparisons are given. Finally, we conclude this study.

## 2. Preliminary

### 2.1. Smart Contract

The smart contract was first proposed by Nick Szabo [19,20] in 1996. A smart contract is a "computer transaction agreement that enforces the terms of the contract". The smart contract allows trusted transactions without a third party. These transactions are traceable and irreversible. The use of smart contract mechanisms reduces the cost of implementing binding agreements between multiple entities, and smart contracts provide a trustworthy commitment.

### 2.2. Proxy Re-Encryption

In 1998, Blaze et al. [21] proposed atomic proxy cryptography, in which a semi-trusted proxy computes a function that converts ciphertexts for Alice into ciphertexts for Bob without seeing the underlying plaintext. In their scheme, with modulus a safe prime $p = 2q + 1$, the proxy is entrusted with the delegation key $b/a$ to change ciphertexts from Alice to Bob via computing ($mg^k \bmod p$; $(g^{ak})^{b/a} \bmod p$), where message $m$ is plaintext and is encrypted with secret keys $a$ and $b$. Proxy cryptography has natural applications to secure file systems. We applied proxy cryptography to secure digital content rights management. Our system uses a centralized digital content administrator and digital content distributor to manage access to encrypted files stored on a distributed, untrusted cloud platform. The digital content administrator manages the digital content resource, and the digital content distributor is the proxy to distribute the digital content under the policy of the museum's authorization. The following scenarios reveal the proxy re-encryption mechanism.

(a) System parameter establishment (key generation)

One of the selectors, a large prime number $p$ and the multiplicative group $Z_p^*$, generates $g$, taking $g$ and $p$ as public parameters. The authorizer Alice randomly selects a positive integer $a < p$ as his/her private key and calculates $g^a \bmod p$. Alice then randomly selects a

positive integer $b < p$ and sends $b$ to the licensee Bob through a secure secret channel as Bob's decryption key, and Bob calculates $g^b$ mod $p$ as the public key.

(b)   Alice encrypts the plaintext $m$

    1.   A positive integer $k$ is selected uniformly and randomly.

    2.   Calculate the ciphertext $(C_1, C_2)$, where $C_1 = mg^k$ mod $p$, $C_2 = g^{ak}$ mod $p$.

(c)   Generation the re-encryption key

If Alice wants to authorize the information to Bob such that Bob can decrypt the ciphertext, Alice sends the ciphertext $(C_1, C_2)$ and proxy key $\pi_{A \to B} = b/a$ to the proxy.

(d)   Re-encryption process

    1.   After receiving the ciphertext $(C_1, C_2) = (mg^k$ mod $p, g^{ak}$ mod $p)$, the proxy uses the re-encryption key $\pi_{A \to B} = b/a$ to re-encrypt $(C_1, C_2)$ into $(C_1', C_2')$.

$$
\begin{aligned}
(C_1', C_2') &= (mg^k \text{ mod } p, g^{ak(\pi_{A \to B})} \text{ mod } p) \\
&= (mg^k \text{ mod } p, g^{ak(b/a)} \text{ mod } p) \\
&= (mg^k \text{ mod } p, g^{bk} \text{ mod } p)
\end{aligned}
$$

    2.   The proxy sends the converted ciphertext $(C_1', C_2') = (mg^k$ mod $p, g^{bk}$ mod $p)$ to Bob.

(e)   Bob decrypts the ciphertext

Bob can decrypt the plaintext $m$ with the multiplicative inverse of $b$:

$$
\frac{C_1'}{C_2'^{(1/b)}} = \frac{mg^k \text{ mod } p}{g^k \text{ mod } p} = m
$$

*2.3. ECDSA*

In the field of cryptography, the Elliptic Curve Digital Signature Algorithm (ECDSA) [22] provides a variant of the standard Digital Signature Algorithm (DSA). Like general elliptic curve cryptography, the bit size of the public key required by ECDSA is about twice the size of the security level. For example, to achieve a security level of 80 bits, the size of the ECDSA public key needs to be 160 bits, and the size of the DSA public key must be at least 1024 bits to achieve the same 80-bit security level.

The signature and verification process of ECDSA is as follows:

Suppose Alice wants to send a message to Bob. Initially, both parties must reach a consensus on the curve parameters $(CURVE, G, n)$. In addition to the field equation of the curve, the base point $G$ on the curve and the multiplication order $n$ of the base point $G$ is also required. Alice also needs a private key $d_A$ and a public key $Q_A$, where $Q_A = d_A G$. If the message Alice wants to send is $m$, Alice needs to choose a random value $k$ between $[1, n-1]$, calculate $z = h(m)$, $(x_1, y_1) = kG$, $r = x_1$ mod $n$, $s = k^{-1}(z + rd_A)$ mod $n$, and send the ECDSA signature pair $(r, s)$ together with the original message $m$ to Bob. After receiving the signature pair $(r, s)$ and the original message $m$, Bob will verify the correctness of the ECDSA signature. Bob first calculates $z\prime = h(m)$, $u_1 = z\prime s^{-1}$ mod $n$, $u_2 = rs^{-1}$ mod $n$, $(x_1\prime, y_1\prime) = u_1 G + u_2 Q_A$, $r \overset{?}{=} x_1\prime$ mod $n$, and if it passes the verification, then Bob confirms that the ECDSA signature and message $m$ sent by Alice are correct.

## 3. Method

*3.1. System Structure*

In this study, we used the Elliptic Curve Digital Signature Algorithm (ECDSA), blockchain, and a smart contract to design a sustainable and traceable cultural relic display mechanism such that the museum social education function of the museum can be carried out at any time. Figure 1 is the system architecture diagram. The roles in the environment include Applicant (A), Blockchain Center (BCC), Digital Content Administrator (DCA), and Digital Content Distributor (DCD).

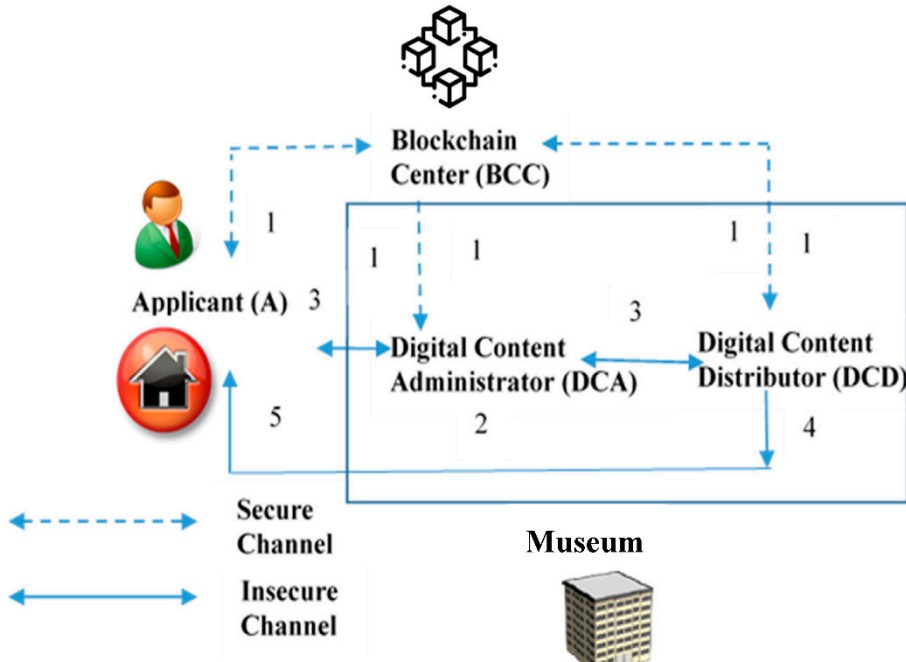

**Figure 1.** System architecture diagram.

(1)  Applicant—It is intended that the citizens or institutions want to access the digital content resource of the museum.
(2)  Blockchain Center (BCC)—This center records the information of the digital rights resource for the Applicant to query or verify the resource status. The BCC accepts the role's registration and issues the identity certificate and public/private key pair to each party.
(3)  Digital Content Administrator (DCA)—The DCA is responsible for the generation and management of the museum's digital content resource. The digital content resource is classified and protected by the DCA. The DCA also reviews the Applicant's request to determine whether to allow or deny access to the digital content resource.
(4)  Digital Content Distributor (DCD)—The DCD is a proxy of the DCA. After the DCA authenticates the Applicant's identification, the DCD is responsible for actual cloud authorization for the Applicant to access the museum's digital content resource. The DCA and DCD are both cloud platforms set up in museums and co-manage the digital content resource.

We briefly illustrate the scenarios in the following steps.

Step 1: Registration phase—Applicant, Digital Content Administrator, and Digital Content Distributor need to register with the Blockchain Center; the Blockchain Center issues the identity certificate and public/private key pair to each party.

Step 2: Digital content production phase—The DCA classifies the museum's resources and then encrypts them into a protected digital resource and then stores it in the DCD. The DCA also uploads the detailed categories into the Blockchain Center.

Step 3: Digital content authorization phase—After the Applicant proposes to access a digital resource request, the DCA reviews the Applicant's qualifications, issues the license key of the Applicant, and transmits the related information to the DCD.

Step 4: Digital content issuing phase—After the DCD receives the DCA's authorization, the DCD uses the proxy re-encryption mechanism to issue the license key, a time-limited key, and player (or reader) to the Applicant.

Step 5: Digital content browsing phase—After the Applicant receives the license key, the Applicant uses it to decrypt the protected digital content. The digital content can be read (or played) normally.

### 3.2. Notation

Table 2 shows the notation of the proposed scheme.

**Table 2.** Notation of the proposed scheme.

| | |
|---|---|
| $q$ | A $k$-bit prime number |
| $GF(q)$ | Finite group $q$ |
| $E$ | The elliptic curve defined on finite group $q$ |
| $G$ | A generating point based on the elliptic curve $E$ |
| $ID_x$ | A name representing identity $x$ |
| $k_x$ | A random value on the elliptic curve |
| $(r_x, s_x)$ | Elliptic curve signature value of $x$ |
| $(x_x, y_x)$ | An ECDSA signature message of $x$ |
| $M_{x\text{-}y}$ | A message sent from x to $y$ |
| $ID_{BC}$ | An index value of blockchain message |
| $BC_x$ | Blockchain message of $x$ |
| $TS_x$ | Timestamp message of $x$ |
| $Enc_x$ | Encrypted message using the asymmetric key of $x$ |
| $ID_{DC}$ | An identity of digital content |
| $key_m$ | A symmetric key containing KeyID and Seed for encrypting/decrypting the digital content |
| $Cert_x$ | A digital certificate of $x$ conforms to the X.509 standard |
| $h(.)$ | Hash function |
| $A \overset{?}{=} B$ | Verify whether $A$ is equal to $B$ |

### 3.3. Registration Phase

The Applicant, DCA, and DCD should register with the Blockchain Center and obtain a relative public/private key pair. The Applicant also gets an identity digital certificate from the Blockchain Center via a secure channel. The system role X can represent the Applicant, DCA, and DCD. Figure 2 shows the flowchart of the registration phase.

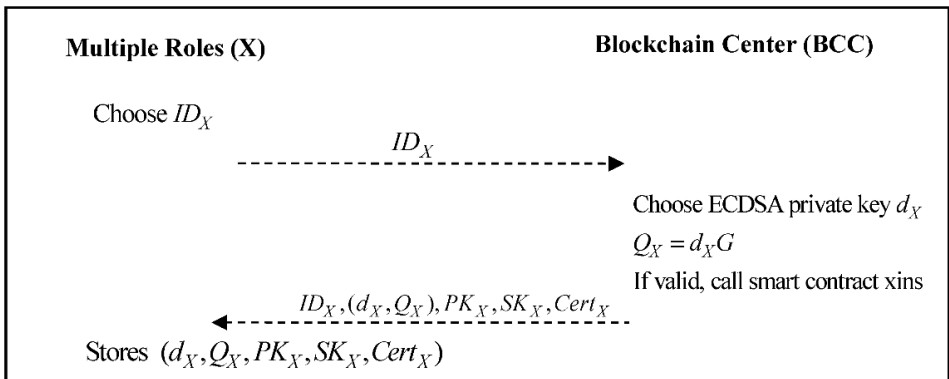

**Figure 2.** Each role of the system registers with the Blockchain Center.

Step 1: System role $X$ generates a name $ID_X$, and sends it to the Blockchain Center.

Step 2: The Blockchain Center generates an ECDSA private key $d_X$ based on the role $X$, calculates $Q_X = d_X G$. If the identity of the registered role is verified, the smart contract xins will be triggered, the content of which is the Algorithm 1 as follows:

---

**Algorithm 1.** Smart contract xins of the proposed scheme.

---

```
function insert x smart contract xins (
string x_id, string x_detail) {
    count ++;
    x[count].id = id;
    x[count].detail = detail;
}
string x_keypairs;
```

---

Then, the Blockchain Center will transmit $ID_X, (d_X, Q_X), PK_X, SK_X, Cert_X$ to role $X$.

Step 3: The role $X$ will store $(d_X, Q_X, PK_X, SK_X, Cert_X)$.

*3.4. Digital Content Production Phase*

The museum collects many precious cultural relics. The digital content production process of valuable cultural relics has a specific process. In general, experts and scholars classify (such as biological classification, antiquities classification, etc.), grade (grade of antiquities is divided into general, important, and national treasures, etc.), and clarify the importance (such as rare or era significance or endangered species, etc.), and then different competent authorities proceed with various kinds of appointments. Finally, it is handed over to professional and technical personnel to produce digital content through photography and 3D surroundings. In this phase, we will focus on illustrating the protection technology of digital content. Figure 3 shows the production flowchart of protected digital content. To enhance performance, we use the digital envelope for implementation. That is, the DCA uses the symmetry key to encrypt the digital content and then employs the ElGamal-based system of the public-key system to protect the symmetry key.

Step 1: The DCA classifies the original multimedia files and encodes them into clob and blob formats for storage. The DCA also makes the identity of these digital contents $ID_{DC}$.

Step 2: The DCA encrypts these encoded multimedia data with KeyID and Seed, and transfers the $key_m$ and encrypted digital content to the DCD for storage.

Step 3: When the Applicant wants to access these multimedia materials, it must first obtain legal authorization from the DCA, and the DCA will provide the Applicant with an authorization key.

Step 4: The Applicant can use the authorization key to unlock the information provided by the DCD and obtain the identity of the digital content $ID_{DC}$. The DCD can then use $ID_{DC}$ to get encryption key keym, which can be used to obtain the plaintext of the digital content. The IDDC must be obtained after authorization verification, and then connect to the DCD from the multimedia player/reader to obtain the keym and play the digital content. This step is performed automatically by the smart contract, and the Applicant cannot skip the verification process privately. The details will be introduced in the rest of the phase.

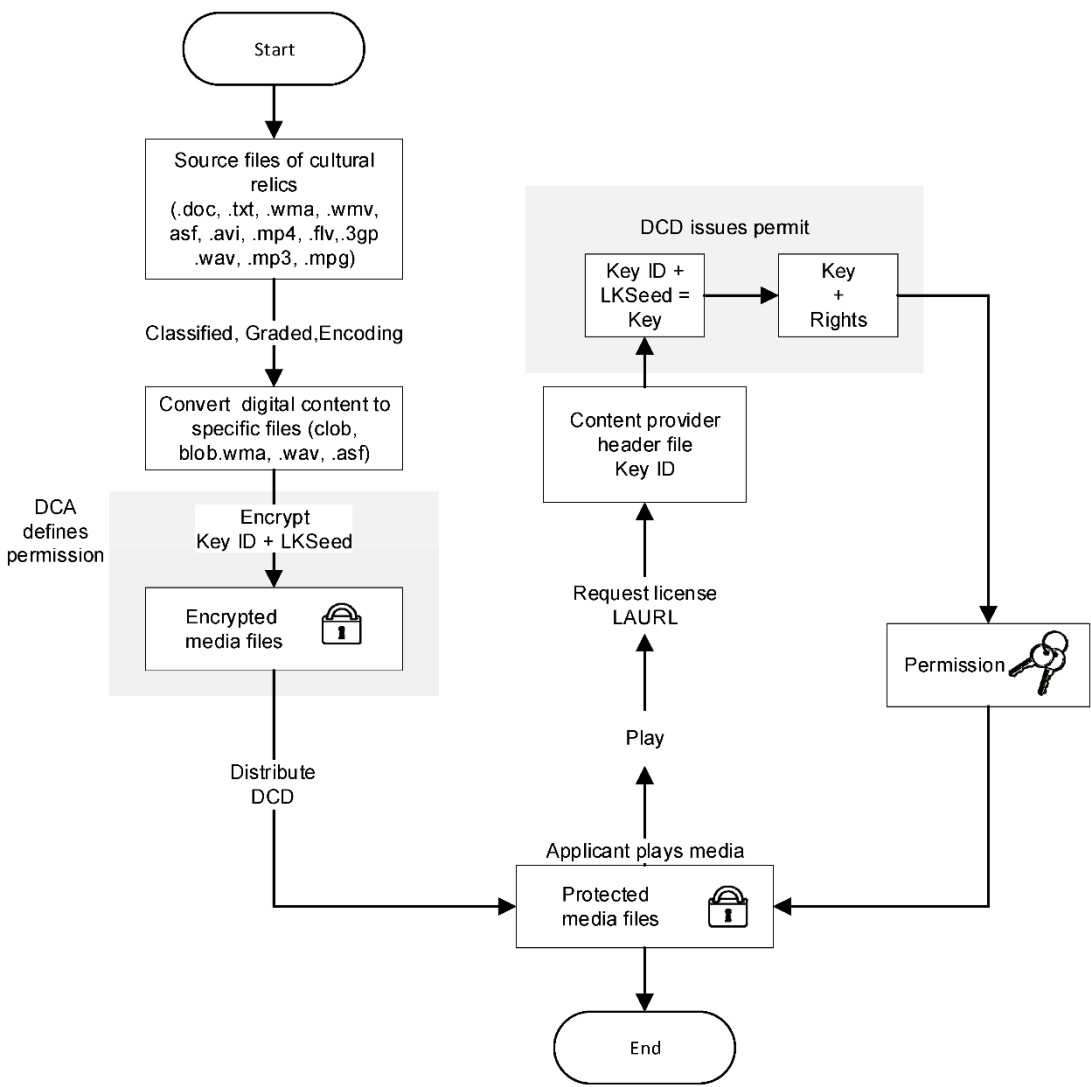

**Figure 3.** The production flowchart of protected digital content.

*3.5. Digital Content Authorization Phase*

After reviewing the Applicant's identity, the DCA, according to the valid authorized time key issues, authorizes access to the digital content. We present the flowchart of the digital content authorization phase in Figure 4.

Step 1: The Applicant generates a random value $k_{A-DCA}$, calculates $z_{A-DCA} = h(ID_A, M_{A-DCA}, Cert_A, TS_{A-DCA}, ID_{BC})$, $(x_{A-DCA}, y_{A-DCA}) = k_{A-DCA}G$, $r_{A-DCA} = x_{A-DCA} \bmod n$, $s_{A-DCA} = k_{A-DCA}^{-1}(z_{A-DCA} + r_{A-DCA}d_A) \bmod n$, $Enc_{A-DCA} = E_{PK_{DCA}}(ID_A, M_{A-DCA}, Cert_A, TS_{A-DCA}, ID_{BC})$, and sends $ID_A, Enc_{A-DCA}, (r_{A-DCA}, s_{A-DCA})$ to the DCA.

Step 2: The DCA first calculates $(ID_A, M_{A-DCA}, Cert_A, TS_{A-DCA}, ID_{BC}) = D_{SK_{DCA}}(Enc_{A-DCA})$, uses $TS_{NOW} - TS_{A-DCA} \leq \Delta T$ to confirm whether the timestamp is valid, and then verifies the correctness of the ECDSA signature, and calculates $z_{A-DCA}' = h(ID_A, M_{A-DCA}, Cert_A, TS_{A-DCA}, ID_{BC})$, $u_{A-DCA1} = z_{A-DCA}'s_{A-DCA}^{-1} \bmod n$, $u_{A-DCA2} = r_{A-DCA}s_{A-DCA}^{-1} \bmod n$, $(x_{A-DCA}', y_{A-DCA}') = u_{A-DCA1}G + u_{A-DCA2}Q_A$, $x_{A-DCA}' \stackrel{?}{=} r_{A-DCA} \bmod n$.

If the verification is passed, the DCA will get the relevant content request information and trigger the smart contracts adcains and adcachk. The content is as follows: timestamp is valid, and then verify the correctness of the ECDSA signature, calculating $z_{A-DCA}' = h(ID_A, M_{A-DCA}, Cert_A, TS_{A-DCA}, ID_{BC})$, $u_{A-DCA1} = z_{A-DCA}'s_{A-DCA}^{-1} \bmod n$, $u_{A-DCA2}$

$= r_{A-DCA}s_{A-DCA}{}^{-1} \bmod n, (x_{A-DCA}{}', y_{A-DCA}{}') = u_{A-DCA1}G + u_{A-DCA2}Q_A, x_{A-DCA}{}' \stackrel{?}{=} r_{A-DCA} \bmod n$.

If the verification is passed, the DCA will get the relevant content request information and trigger the smart contracts adcains and adcachk. The content is the Algorithm 2 as follows:

---

**Algorithm 2.** Smart contract adcains and adcachk of the proposed scheme.

---

```
function insert smart contract adcains(
string adca_id, string adca_detail,
string adca_cert, string adca_tsp) {
     count ++;
     adca[count].id = id;
     adca[count].detail = detail;
     adca[count].cert = cert;
     adca[count].tsp = tsp;
}
sign string a_key (adca_id, adca_detail,
adca_cert, adca_tsp);
verify string a_key (adca_id, adca_detail,
adca_cert, adca_tsp);
function check smart contract adcachk(
string adca_id, string adca_detail,
string adca_cert, string adca_tsp) {
     return adca_id.exist;
     return adca_detail.exist;
     return adca_cert.exist;
   return adca_tsp.exist;
}
```

---

The DCA calculates $BC_{A-DCA} = h(r_{A-DCA}, s_{A-DCA})$; $(ID_{BC}, BC_{A-DCA})$ will also be uploaded to the Blockchain Center. Then the DCA generates a random value $k_{DCA-A}$ and calculates $b = h(Cert_A, Vtime_A)$, $z_{DCA-A} = h(ID_{DCA}, M_{DCA-A}, b, TS_{DCA-A}, BC_{A-DCA}, ID_{BC})$, $(x_{DCA-A}, y_{DCA-A}) = k_{DCA-A}G$, $r_{DCA-A} = x_{DCA-A} \bmod n$, $s_{DCA-A} = k_{DCA-A}{}^{-1}$ $(z_{DCA-A} + r_{DCA-A}d_{DCA}) \bmod n$, $Enc_{DCA-A} = E_{PK_A}(ID_{DCA}, M_{DCA-A}, b, TS_{DCA-A}, BC_{A-DCA}, ID_{BC})$, and sends $ID_{DCA}, Enc_{DCA-A}, (r_{DCA-A}, s_{DCA-A})$ to the Applicant.

Step 3: The Applicant first calculates $(ID_{DCA}, M_{DCA-A}, b, TS_{DCA-A}, BC_{A-DCA}, ID_{BC})$ $= D_{SK_{DCA}}(Enc_{DCA-A})$, uses $TS_{NOW} - TS_{DCA-A} \leq \Delta T$ to confirm whether the timestamp is valid, then verifies the correctness of the ECDSA signature, and calculates $z_{DCA-A}{}' = h(IID_{DCA}, M_{DCA-A}, b, TS_{DCA-A}, BC_{A-DCA}, ID_{BC})$, $u_{DCA-A1} = z_{DCA-A}{}'s_{DCA-A}{}^{-1} \bmod n$, $u_{DCA-A2} = r_{DCA-A}s_{DCA-A}{}^{-1} \bmod n$, $(x_{DCA-A}{}', y_{DCA-A}{}') = u_{DCA-A1}G + u_{DCA-A2}Q_{DCA}$, $x_{DCA-A}{}' \stackrel{?}{=} r_{DCA-A} \bmod n$.

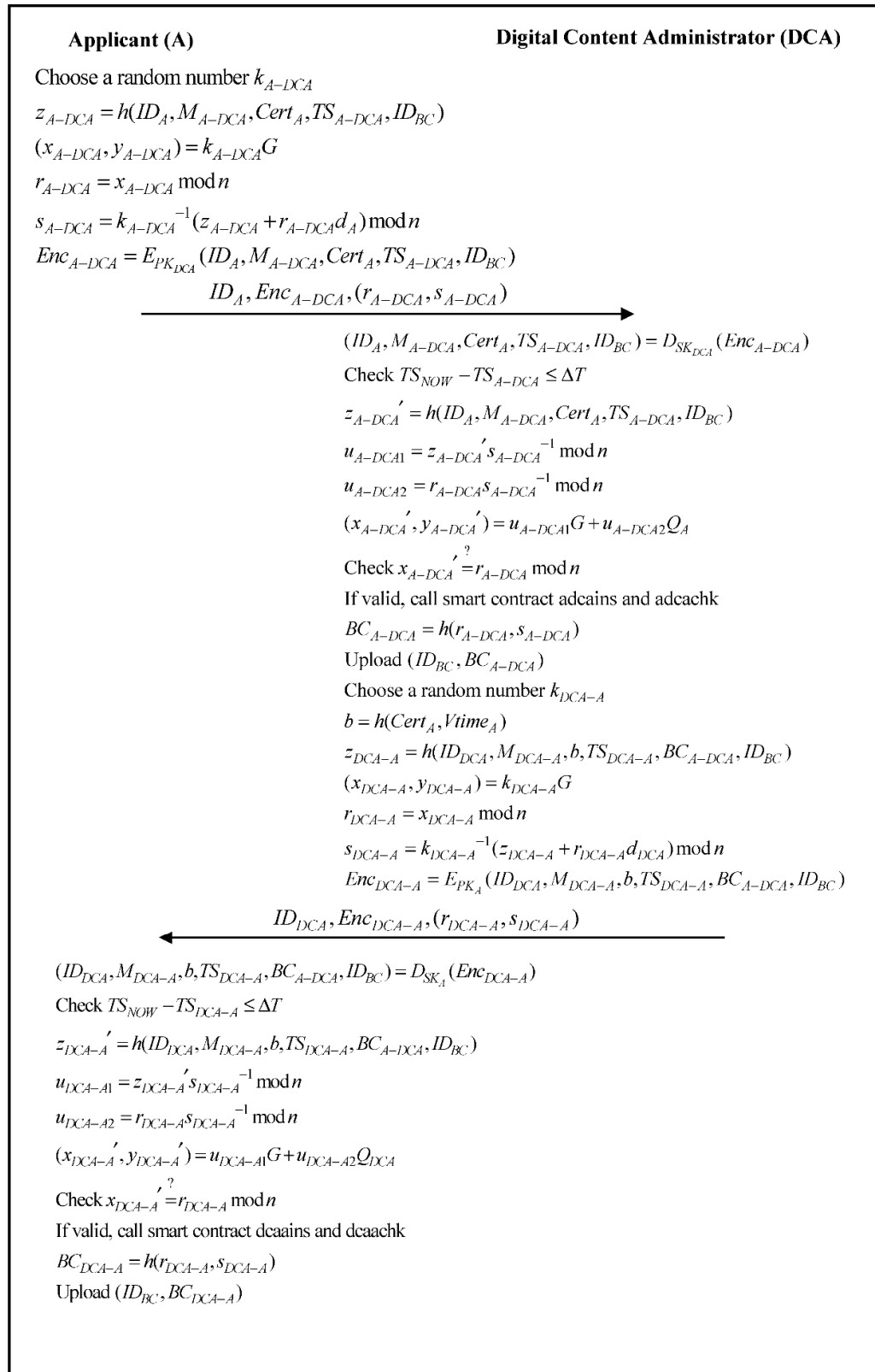

**Figure 4.** Digital content authorization phase.

If the verification is passed, the content request information is confirmed by the DCA, and the smart contracts dcaains and dcaachk will be sent. The content is the Algorithm 3 as follows:

---

**Algorithm 3.** Smart contract dcaains and dcaachk of the proposed scheme.

---

```
function insert smart contract dcaains(
string dcaa_id, string dcaa_detail,
string dcaa_akey, string dcaa_tsp) {
    count ++;
    dcaa[count].id = id;
    dcaa[count].detail = detail;
    dcaa[count].akey = akey;
    dcaa[count].tsp = tsp;
}
sign string dca_key (dcaa_id, dcaa_detail,
dcaa_akey, dcaa_tsp);
verify string dca_key (dcaa_id, dcaa_detail,
dcaa_akey, dcaa_tsp);
function check smart contract dcaachk(
string dcaa_id, string dcaa_detail,
string dcaa_akey, string dcaa_tsp) {
    return dcaa_id.exist;
    return dcaa_detail.exist;
    return dcaa_akey.exist;
    return dcaa_tsp.exist;
}
```

---

The Applicant calculates $BC_{DCA-A} = h(r_{DCA-A}, s_{DCA-A})$, and $(ID_{BC}, BC_{DCA-A})$ will also be uploaded to the Blockchain Center.

*3.6. Digital Content Issuing Phase*

The DCD is to be a proxy and uses the proxy re-encryption mechanism to issue the license key—time-limited key and player (or reader) to the Applicant. Figure 5 illustrates the flowchart of the digital content issuing phase.

Step 1: The DCA generates a random value $k_{DCA-DCD}$, calculates $key_m = (KeyID, Seed)$, $c_1 = ID_{DC}g^{ak} \bmod p$, $c_2 = g^k \bmod p$, $z_{DCA-DCD} = h(ID_{DCA}, M_{DCA-DCD}, c_1, c_2, b/a, key_m, TS_{DCA-DCD}, ID_{BC})$, $(x_{DCA-DCD}, y_{DCA-DCD}) = k_{DCA-DCD}G$, $r_{DCA-DCD} = x_{DCA-DCD} \bmod n$, $s_{DCA-DCD} = k_{DCA-DCD}^{-1}(z_{DCA-DCD} + r_{DCA-DCD}d_{DCA}) \bmod n Enc_{DCA-DCD} = E_{PK_{DCD}} (ID_{DCA}, M_{DCA-DCD}, c_1, c_2, b/a, key_m, TS_{DCA-DCD}, ID_{BC})$, and sends $ID_{DCA}$, $Enc_{DCA-DCD}$, $(r_{DCA-DCD}, s_{DCA-DCD})$ to the DCD.

Step 2: The DCD first calculates $(ID_{DCA}, M_{DCA-DCD}, c_1, c_2, b/a, key_m, TS_{DCA-DCD}, ID_{BC}) = D_{SK_{DCD}}(Enc_{DCA-DCD})$, uses $TS_{NOW} - TS_{DCA-DCD} \leq \Delta T$ to confirm whether the timestamp is valid, then verifies the correctness of the ECDSA signature, and calculates $z_{DCA-DCA}' = h(ID_{DCA}, M_{DCA-DCD}, c_1, c_2, b/a, key_m, TS_{DCA-DCD}, ID_{BC})$, $u_{DCA-DCD1} = z_{DCA-DCD}'s_{DCA-DCD}^{-1} \bmod n$, $u_{DCA-DCD2} = r_{DCA-DCD}s_{DCA-DCD}^{-1} \bmod n$, $(x_{DCA-DCD}', y_{DCA-DCD}') = u_{DCA-DCD1}G + u_{DCA-DCD2}Q_{DCA}$, $x_{DCA-DCD}' \stackrel{?}{=} r_{DCA-DCD} \bmod n$.

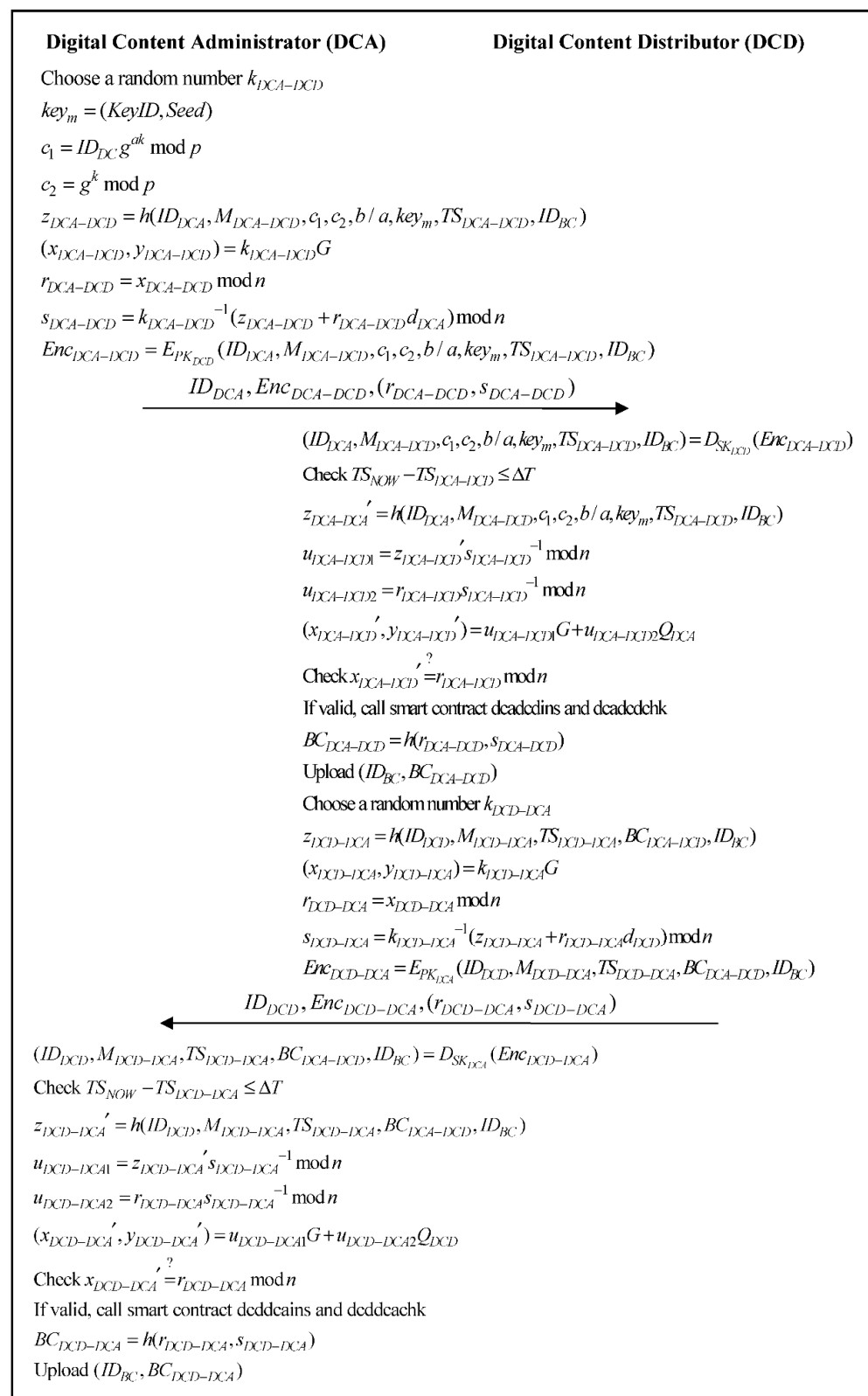

**Figure 5.** Digital content issuing phase.

If the verification is passed, the DCD will get the relevant content issuing information and trigger the smart contracts dcadcdins and dcadcdchk. The content is the Algorithm 4 as follows:

---

**Algorithm 4.** Smart contract dcadcdins and dcadcdchk of the proposed scheme.

---

function insert smart contract dcadcdins(
string dcadcd_id, string dcadcd_detail,
string dcadcd_iddc, string dcadcd_ekey,
string dcadcd_akey, string dcadcd_tsp) {
    count ++;
    dcadcd[count].id = id;
    dcadcd[count].detail = detail;
    dcadcd[count].iddc = iddc;
    dcadcd[count].ekey = ekey;
    dcadcd[count].akey = akey;
    dcadcd[count].tsp = tsp;
}
sign string dca_key (dcadcd_id, dcadcd_detail,
dcadcd_iddc, dcadcd_ekey, dcadcd_akey,
dcadcd_tsp);
verify string dca_key (dcadcd_id, dcadcd_detail, dcadcd_iddc, dcadcd_ekey, dcadcd_akey,
dcadcd_tsp);
function check smart contract dcadcdchk(
string dcadcd_id, string dcadcd_detail,
string dcadcd_iddc, string dcadcd_ekey,
string dcadcd_akey, string dcadcd_tsp) {
    return dcadcd_id.exist;
    return dcadcd_detail.exist;
    return dcadcd_iddc.exist;
    return dcadcd_ekey.exist;
    return dcadcd_akey.exist;
    return dcadcd_tsp.exist;
}

---

The DCD calculates $BC_{DCA-DCD} = h(r_{DCA-DCD}, s_{DCA-DCD})$; $(ID_{BC}, BC_{DCA-DCD})$ will also be uploaded to the Blockchain Center. The DCD then generates a random value $k_{DCD-DCA}$ and calculates $z_{DCD-DCA} = h(ID_{DCD}, M_{DCD-DCA}, TS_{DCD-DCA}, BC_{DCA-DCD}, ID_{BC})$, $(x_{DCD-DCA}, y_{DCD-DCA}) = k_{DCD-DCA}G$, $r_{DCD-DCA} = x_{DCD-DCA} \bmod n$, $s_{DCD-DCA} = k_{DCD-DCA}{}^{-1}(z_{DCD-DCA} + r_{DCD-DCA}d_{DCD}) \bmod n Enc_{DCD-DCA} = E_{PK_{DCA}}$ $(ID_{DCD}, M_{DCD-DCA}, TS_{DCD-DCA}, BC_{DCA-DCD}, ID_{BC})$, and sends $ID_{DCD}, Enc_{DCD-DCA}$, $(r_{DCD-DCA}, s_{DCD-DCA})$ to the DCA.

Step 3: The DCA first calculates $(ID_{DCD}, M_{DCD-DCA}, TS_{DCD-DCA}, BC_{DCA-DCD}, ID_{BC})$ $= D_{SK_{DCA}}(Enc_{DCD-DCA})$, uses $TS_{NOW} - TS_{DCD-DCA} \leq \Delta T$ to confirm whether the timestamp is valid, then verifies the correctness of the ECDSA signature, and calculates $z_{DCD-DCA}' = h(ID_{DCD}, M_{DCD-DCA}, TS_{DCD-DCA}, BC_{DCA-DCD}, ID_{BC})$, $u_{DCD-DCA1} = z_{DCD-DCA}'s_{DCD-DCA}{}^{-1} \bmod n$, $u_{DCD-DCA2} = r_{DCD-DCA}s_{DCD-DCA}{}^{-1} \bmod n$, $(x_{DCD-DCA}', y_{DCD-DCA}') = u_{DCD-DCA1}G + u_{DCD-DCA2}Q_{DCD}$, $x_{DCD-DCA}' \overset{?}{=} r_{DCD-DCA}$ $\bmod n$. If the verification is passed, the content issuing information is confirmed by the DCD, and the smart contracts dcddcains and dcddcachk will be sent. The content is the Algorithm 5 as follows:

**Algorithm 5.** Smart contract dcddcains and dcddcachk of the proposed scheme.

```
function insert smart contract dcddcains(
string dcddca_id, string dcddca_detail,
string dcddca_tsp) {
    count ++;
    dcddca[count].id = id;
    dcddca[count].detail = detail;
    dcddca[count].tsp = tsp;
}
sign string dcd_key (dcddca_id, dcddca_detail,
dcddca_tsp);
verify string dcd_key (dcddca_id, dcddca_detail,
dcddca_tsp);
function check smart contract dcddcachk(
string dcddca_id, string dcddca_detail,
string dcddca_tsp) {
    return dcddca_id.exist;
    return dcddca_detail.exist;
    return dcddca_tsp.exist;
}
```

The DCA calculates $BC_{DCD-DCA} = h(r_{DCD-DCA}, s_{DCD-DCA})$; $(ID_{BC}, BC_{DCD-DCA})$ will also be uploaded to the Blockchain Center.

*3.7. Digital Content Browsing Phase*

After authorization, the Applicant's application (reader or player) can use the authorized key to decrypt the symmetry key. The Applicant can browse the digital content normally. Figure 6 illustrates the flowchart of the digital content browsing phase.

Step 1: The Applicant downloads the application and generates a random value $k_{A-DCD}$, calculates $z_{A-DCD} = h(ID_A, M_{A-DCD}, TS_{A-DCD}, ID_{BC})$, $(x_{A-DCD}, y_{A-DCD}) = k_{A-DCD}G$, $r_{A-DCD} = x_{A-DCD} \bmod n s_{A-DCD} = k_{A-DCD}^{-1}(z_{A-DCD} + r_{A-DCD}d_A) \bmod n$, $Enc_{A-DCD} = E_{PK_{DCD}}(ID_A, M_{A-DCD}, TS_{A-DCD}, ID_{BC})$, and sends $ID_A, Enc_{A-DCD}$, $(r_{A-DCD}, s_{A-DCD})$ to the DCD.

Step 2: The DCD first calculates $(ID_A, M_{A-DCD}, TS_{A-DCD}, ID_{BC}) = D_{SK_{DCD}(Enc_{A-DCD})}$, uses $TS_{NOW} - TS_{A-DCD} \leq \Delta T$ to confirm whether the timestamp is valid, and then verifies the correctness of the ECDSA signature, calculates $z_{A-DCD}' = h(ID_A, M_{A-DCD}, TS_{A-DCD}, ID_{BC})$, $u_{A-DCD1} = z_{A-DCD}'s_{A-DCD}^{-1} \bmod n$, $u_{A-DCD2} = r_{A-DCD}s_{A-DCD}^{-1} \bmod n$, $(x_{A-DCD}', y_{A-DCD}') = u_{A-DCD1}G + u_{A-DCD2}Q_A$, $x_{A-DCD}' \overset{?}{=} r_{A-DCD} \bmod n$.

**Applicant (A)**                                    **Digital Content Distributor (DCD)**

Choose a random number $k_{A-DCD}$

$z_{A-DCD} = h(ID_A, M_{A-DCD}, TS_{A-DCD}, ID_{BC})$

$(x_{A-DCD}, y_{A-DCD}) = k_{A-DCD}G$

$r_{A-DCD} = x_{A-DCD} \bmod n$

$s_{A-DCD} = k_{A-DCD}^{-1}(z_{A-DCD} + r_{A-DCD}d_A) \bmod n$

$Enc_{A-DCD} = E_{PK_{DCD}}(ID_A, M_{A-DCD}, TS_{A-DCD}, ID_{BC})$

$$ID_A, Enc_{A-DCD}, (r_{A-DCD}, s_{A-DCD}) \longrightarrow$$

$(ID_A, M_{A-DCD}, TS_{A-DCD}, ID_{BC}) = D_{SK_{DCD}}(Enc_{A-DCD})$

Check $TS_{NOW} - TS_{A-DCD} \leq \Delta T$

$z_{A-DCD}' = h(ID_A, M_{A-DCD}, TS_{A-DCD}, ID_{BC})$

$u_{A-DCD1} = z_{A-DCD}' s_{A-DCD}^{-1} \bmod n$

$u_{A-DCD2} = r_{A-DCD} s_{A-DCD}^{-1} \bmod n$

$(x_{A-DCD}', y_{A-DCD}') = u_{A-DCD1}G + u_{A-DCD2}Q_A$

Check $x_{A-DCD}' \overset{?}{=} r_{A-DCD} \bmod n$

If valid, call smart contract adcdins and adcdchk

$BC_{A-DCD} = h(r_{A-DCD}, s_{A-DCD})$

Upload $(ID_{BC}, BC_{A-DCD})$

Choose a random number $k_{DCD-A}$

$c_1' = c_1^{(b/a)}$

$z_{DCD-A} = h(ID_{DCD}, M_{DCD-A}, c_1', c_2, TS_{DCD-A}, BC_{A-DCD}, ID_{BC})$

$(x_{DCD-A}, y_{DCD-A}) = k_{DCD-A}G$

$r_{DCD-A} = x_{DCD-A} \bmod n$

$s_{DCD-A} = k_{DCD-A}^{-1}(z_{DCD-A} + r_{DCD-A}d_{DCD}) \bmod n$

$Enc_{DCD-A} = E_{PK_A}(ID_{DCD}, M_{DCD-A}, c_1', c_2, TS_{DCD-A}, BC_{A-DCD}, ID_{BC})$

$$\longleftarrow ID_{DCD}, Enc_{DCD-A}, (r_{DCD-A}, s_{DCD-A})$$

$(ID_{DCD}, M_{DCD-A}, c_1', c_2, TS_{DCD-A}, BC_{A-DCD}, ID_{BC}) = D_{SK_A}(Enc_{DCD-A})$

Check $TS_{NOW} - TS_{DCD-A} \leq \Delta T$

$z_{DCD-A}' = h(ID_{DCD}, M_{DCD-A}, c_1', c_2, TS_{DCD-A}, BC_{A-DCD}, ID_{BC})$

$u_{DCD-A1} = z_{DCD-A}' s_{DCD-A}^{-1} \bmod n$

$u_{DCD-A2} = r_{DCD-A} s_{DCD-A}^{-1} \bmod n$

$(x_{DCD-A}', y_{DCD-A}') = u_{DCD-A1}G + u_{DCD-A2}Q_{DCD}$

Check $x_{DCD-A}' \overset{?}{=} r_{DCD-A} \bmod n$

If valid, call smart contract dcdains and dcdachk

$BC_{DCD-A} = h(r_{DCD-A}, s_{DCD-A})$

Upload $(ID_{BC}, BC_{DCD-A})$

$ID_{DC} = c_1'^{(b)} / c_2$

Browse the digital content through the reader or player

**Figure 6.** Digital content browsing phase.

If it passes the verification, the DCD will get the relevant content and trigger the smart contracts adcdins and adcdchk. The content is the Algorithm 6 as follows:

---

**Algorithm 6.** Smart contract adcdins and adcdchk of the proposed scheme.

---

```
function insert smart contract adcdins(
string adcd_id, string adcd_detail,
string adcd_tsp) {
    count ++;
    adcd[count].id = id;
    adcd[count].detail = detail;
    adcd[count].tsp = tsp;
}
sign string a_key (adcd_id, adcd_detail,
adcd_tsp);
function insert smart contract adcdins(
string adcd_id, string adcd_detail,
string adcd_tsp) {
    count ++;
    adcd[count].id = id;
    adcd[count].detail = detail;
    adcd[count].tsp = tsp;
}
sign string a_key (adcd_id, adcd_detail,
adcd_tsp);
```

---

The DCD calculates $BC_{A-DCD} = h(r_{A-DCD}, s_{A-DCD})$; $(ID_{BC}, BC_{A-DCD})$ will also be uploaded to the Blockchain Center. The DCD then generates a random value $k_{DCD-A}$ and calculates $c_1' = c_1^{(b/a)}$, $z_{DCD-A} = h(ID_{DCD}, M_{DCD-A}, c_1', c_2, TS_{DCD-A}, BC_{A-DCD}, ID_{BC})$ $(x_{DCD-A}, y_{DCD-A}) = k_{DCD-A}G$, $r_{DCD-A} = x_{DCD-A} \bmod n$, $s_{DCD-A} = k_{DCD-A}^{-1}$ $(z_{DCD-A} + r_{DCD-A}d_{DCD}) \bmod n$,

$Enc_{DCD-A} = E_{PK_A}(ID_{DCD}, M_{DCD-A}, c_1', c_2, TS_{DCD-A}, BC_{A-DCD}, ID_{BC})$, and sends $ID_{DCD}, Enc_{DCD-A}, (r_{DCD-A}, s_{DCD-A})$ to the Applicant.

Step 3: The Applicant's application first calculates $(ID_{DCD}, M_{DCD-A}, c_1', c_2, TS_{DCD-A}, BC_{A-DCD}, ID_{BC}) = D_{SK_A}(Enc_{DCD-A})$, uses $TS_{NOW} - TS_{DCD-A} \leq \Delta T$ to confirm whether the timestamp is valid, then verifies the correctness of the ECDSA signature, and calculates $z_{DCD-A}' = h(ID_{DCD}, M_{DCD-A}, c_1', c_2, TS_{DCD-A}, BC_{A-DCD}, ID_{BC})$, $u_{DCD-A1} = z_{DCD-A}'s_{DCD-A}^{-1} \bmod n$, $u_{DCD-A2} = r_{DCD-A}s_{DCD-A}^{-1} \bmod n$, $(x_{DCD-A}', y_{DCD-A}') = u_{DCD-A1}G + u_{DCD-A2}Q_{DCD}$, $x_{DCD-A}' \overset{?}{=} r_{DCD-A} \bmod n$. If it passes the verification, the content browsing request information is confirmed by the DCD, and the smart contracts dcdains and dcdachk will be executed. The content is the Algorithm 7 as follows:

---

**Algorithm 7.** Smart contract dcdains and dcdachk of the proposed scheme.

---

```
function insert smart contract dcdains(
string dcda_id, string dcda_detail,
string dcda_akey, string dcda_iddc,
string dcda_tsp) {
    count ++;
    dcda[count].id = id;
    dcda[count].detail = detail;
    dcda[count].akey = akey;
    dcda[count].iddc = iddc;
    dcda[count].tsp = tsp;
}
sign string dcd_key (dcda_id, dcda_detail,
dcda_akey, dcda_iddc, dcda_tsp);
verify string dcd_key (dcda_id, dcda_detail,
dcda_akey, dcda_iddc, dcda_tsp);
function check smart contract dcdachk(
string dcda_id, string dcda_detail,
string dcda_akey, string dcda_iddc,
string dcda_tsp) {
    return dcda_id.exist;
    return dcda_detail.exist;
    return dcda_akey.exist;
    return dcda_iddc.exist;
    return dcda_tsp.exist;
}
```

---

The Applicant calculates $BC_{DCD-A} = h(r_{DCD-A}, s_{DCD-A})$; $(ID_{BC}, BC_{DCD-A})$ will also be uploaded to the Blockchain Center. Finally, the Applicant's application calculates $ID_{DC} = c_1'^{(b)}/c_2$ to obtain the identity of the digital content $ID_{DC}$ successfully. Therefore, the DCD can use $ID_{DC}$ to obtain the corresponding decryption key $key_m$, which can be used to obtain the plaintext of digital content through the reader or player. This step is performed automatically by the smart contract, and the Applicant cannot skip the verification process privately.

## 4. Analysis

### 4.1. Public Authentication

Using digital certificate verification can publicly verify the identity of the applicant, and the authorization information can be made public. The open transparency of the museum's information on the chain solves the actual problematic points of information asymmetry and cumbersome process and low efficiency, breaking the original barriers, and truly realizing high efficiency and specialization in the field of museum's digital rights. Let us take the message transmitted by the Applicant (A) and Digital Content Administrator (DCA) as an example. When the DCA sends a message signed by ECDSA to the Applicant, the Applicant will first verify the correctness of the time stamp and signature, then generate blockchain data $BC_{DCA-A} = h(r_{DCA-A}, s_{DCA-A})$, and use $ID_{BC}$ as an index to upload the blockchain data to the Blockchain Center (BCC). That is to say, after verifying the correctness of the time stamp and signature for each role that receives the message, it also verifies the correctness of the blockchain data generated by the previous role. Therefore, our proposed solution achieves the characteristics of public verification through blockchain technology and ECDSA digital signature.

### 4.2. Transparency

The identity of the authorized object of digital content is verified by the Digital Content Administrator of the museum. The authorization period is controlled by the Digital

Content Administrator. The Applicant cannot occupy or transfer privately. Any nodes that participate in the system do not need to trust each other. The operation of the system and operating rules are open and transparent, and all information is open. A node cannot deceive other nodes. In this way, the trust relationship between nodes is realized, making it possible to obtain trust between nodes at a low cost. For example, when the Applicant (A) requests digital content authorization from the Digital Content Administrator (DCA), the DCA will send an authorization key to A. This key $b = h(Cert_A, Vtime_A)$ contains A's certificate and the hash value of the authorization period; A will not be able to privately occupy or transfer digital content privately.

### 4.3. Tamperproof

Use the timestamp and signature mechanism to irreversibly generate a string composed of random numbers and letters for the data placed in each block. This original text cannot be inferred from the string, thus effectively solving the trust problem. The messages are illustrated as follows after performing the hash function operation:

$z_{A-DCA} = h(ID_A, M_{A-DCA}, Cert_A, TS_{A-DCA}, ID_{BC})$
$z_{DCA-A} = h(ID_{DCA}, M_{DCA-A}, b, TS_{DCA-A}, BC_{A-DCA}, ID_{BC})$
$z_{DCA-DCD} = h(ID_{DCA}, M_{DCA-DCD}, c_1, c_2, b/a, key_m, TS_{DCA-DCD}, ID_{BC})$
$z_{DCD-DCA} = h(ID_{DCD}, M_{DCD-DCA}, TS_{DCD-DCA}, BC_{DCA-DCD}, ID_{BC})$
$z_{A-DCD} = h(ID_A, M_{A-DCD}, TS_{A-DCD}, ID_{BC})$
$z_{DCD-A} = h(ID_{DCD}, M_{DCD-A}, c_1', c_2, TS_{DCD-A}, BC_{A-DCD}, ID_{BC})$

The messages cannot be reversed back to the original content, so this agreement achieves the characteristic that the message cannot be tampered with.

### 4.4. Traceable

After the digital content of the museum is on the chain, the data block containing the copyright information is permanently stored on the blockchain and cannot be tampered with. All transaction traces can be traced throughout the entire process, which can be used as a digital deposit certificate of the museum to deal with infringement. For example, when we want to verify and trace whether the blockchain data between the Applicant (A) and Digital Content Administrator (DCA) is legal, we can compare and verify $BC_{A-DCA} \overset{?}{=} h(r_{A-DCA}, s_{A-DCA})$ and $BC_{DCA-A} \overset{?}{=} h(r_{DCA-A}, s_{DCA-A})$.

When we want to verify and trace whether the blockchain data between the Digital Content Administrator (DCA) and Digital Content Distributor (DCD) is legal, we can compare and verify $BC_{DCA-DCD} \overset{?}{=} h(r_{DCA-DCD}, s_{DCA-DCD})$ and $BC_{DCD-DCA} \overset{?}{=} h(r_{DCD-DCA}, s_{DCD-DCA})$. When we want to verify and trace whether the blockchain data between the Applicant (A) and Digital Content Distributor (DCD) is legal, we can compare and verify $BC_{A-DCD} \overset{?}{=} h(r_{A-DCD}, s_{A-DCD})$ and $BC_{DCD-A} \overset{?}{=} h(r_{DCD-A}, s_{DCD-A})$.

### 4.5. Nonrepudiation

The content of the message sent by each role is signed by the role of its ECDSA private key. After the receiver receives the message, it will verify the message with the sender's public key. If the message is successfully verified, the sender cannot deny the content of the message is transmitted. Table 3 is an undeniable description of each role in this program.

**Table 3.** Non-repudiation of the proposed scheme.

| Phase / Item | Signature Value | Sender | Receiver | Signature Verification |
|---|---|---|---|---|
| Digital content authorization phase | $(r_{A-DCA}, s_{A-DCA})$ | A | DCA | $x_{A-DCA}' \overset{?}{=} r_{A-DCA} \bmod n$ |
| | $(r_{DCA-A}, s_{DCA-A})$ | DCA | A | $x_{DCA-A}' \overset{?}{=} r_{DCA-A} \bmod n$ |
| Digital content issuing phase | $(r_{DCA-DCD}, s_{DCA-DCD})$ | DCA | DCD | $x_{DCA-DCD}' \overset{?}{=} r_{DCA-DCD} \bmod n$ |
| | $(r_{DCD-DCA}, s_{DCD-DCA})$ | DCD | DCA | $x_{DCD-DCA}' \overset{?}{=} r_{DCD-DCA} \bmod n$ |
| Digital content browsing phase | $(r_{A-DCD}, s_{A-DCD})$ | A | DCD | $x_{A-DCD}' \overset{?}{=} r_{A-DCD} \bmod n$ |
| | $(r_{DCD-A}, s_{DCD-A})$ | DCD | A | $x_{DCD-A}' \overset{?}{=} r_{DCD-A} \bmod n$ |

### 4.6. Data Format Standardization

Effectively categorizing the digital content of the museum and formatting it on the chain is helpful to effectively manage digital property rights and control the unique authorization power of digital content, and intellectual property rights can be protected.

This study re-encodes various original multimedia files into blob and clob formats, which will provide fast and consistent authorized content transmission services.

### 4.7. Timeless

In our proposed scheme, the Digital Content Administrator (DCA) of the museum is responsible for the production and management of the digital content property rights and the identity verification of the Applicant (A); the Digital Content Administrator (DCA) is also responsible for the issuance of a time-sensitive playback license, and the applicant's playback key identification code cannot permanently occupy the playback of digital content. The Applicant must obtain the decryption key through the authorization key; the authorization key contains the hash value of the Applicant's certificate and the authorization period. If the authorization period expires, the Applicant will not be able to obtain the decryption key; that is, it cannot perform digital content playback. Thus, we will not worry about the leakage of digital property rights.

## 5. Discussion, Comparisons, and Limitations

### 5.1. Computation Cost

Table 4 is the computation cost analysis of all stages and roles in this scheme. We analyzed the digital content issuing phase with the highest computational cost. The DCA requires eight multiplication operations, three hash function operations, two comparison operations, and three signature operations. The DCD requires seven multiplication operations, three hash function operations, two comparison operations, and three signature operations in the digital content issuing phase. The proposed scheme has a good computational cost.

**Table 4.** Computation cost analysis of this scheme.

| Phase \ Role | BCC | A | DCA | DCD |
|---|---|---|---|---|
| System role registration phase | $1T_{Mul}$ | N/A | N/A | N/A |
| Digital content authorization phase | N/A | $7T_{Mul} + 3T_H$ $+2T_{Cmp} + 2T_{Sig}$ | $7T_{Mul} + 4T_H$ $+2T_{Cmp} + 2T_{Sig}$ | N/A |
| Digital content issuing phase | N/A | N/A | $8T_{Mul} + 3T_H$ $+2T_{Cmp} + 3T_{Sig}$ | $7T_{Mul} + 3T_H$ $+2T_{Cmp} + 3T_{Sig}$ |
| Digital content browsing phase | N/A | $7T_{Mul} + 3T_H$ $+2T_{Cmp} + 2T_{Sig}$ | N/A | $7T_{Mul} + 3T_H$ $+2T_{Cmp} + 2T_{Sig}$ |

$T_{Mul}$: Multiplication operation; $T_H$: Hash function operation; $T_{Cmp}$: Comparison operation; $T_{sig}$: Signature operation.

## 5.2. Communication Cost

The communication cost analysis of each phase in this scheme is shown in Table 5.

**Table 5.** Communication cost analysis of this scheme.

| Phase \ Item | Message Length | Rounds | 3.5 G (14 Mbps) | 4 G (100 Mbps) | 5 G (20 Gbps) |
|---|---|---|---|---|---|
| System role registration phase | 3552 bits | 2 | 0.254 ms | 0.036 ms | 0.178 us |
| Digital content authorization phase | 2848 bits | 2 | 0.203 ms | 0.028 ms | 0.142 us |
| Digital content issuing phase | 2848 bits | 2 | 0.203 ms | 0.028 ms | 0.142 us |
| Digital content browsing phase | 2848 bits | 2 | 0.203 ms | 0.028 ms | 0.142 us |

We assumed that the ECDSA key and signature were 160 bits, the asymmetric message or certificate was 1024 bits, and the rest of the message length, such as ID, was 80 bits. We analyzed the system role registration phase with the highest communication cost. The message sent by the system role to the Blockchain Center includes one other message. The message sent by the Blockchain Center to the system role includes two ECDSA keys and signatures, three asymmetric messages or certificates, and one other message. The total communication cost in the system role registration phase is 3552 bits, which takes 0.254 ms under a 3.5 G (14 Mbps) communication environment, 0.036 ms under a 4 G (100 Mbps) communication environment, and takes 0.178 us under a 5 G (20 Mbps) communication environment [23]. The proposed scheme has excellent performance.

## 5.3. Comparison

We involved state-of-the-art technology to protect valuable cultural relics of the museum and designed a realistic authorization scheme. In this subsection, we compared museum DRM [3,4] and detailed DRM protocols [5–7,14–17]. Table 6 is the comparison between the proposed and existing digital rights management surveys.

**Table 6.** Comparison of the proposed and existing digital rights management surveys.

| Authors | Year | Objective | 1 | 2 | 3 | 4 | 5 | 6 | 7 | 8 |
|---|---|---|---|---|---|---|---|---|---|---|
| Triet and Due [3] | 2004 | Proposed an application of the psychoacoustic audio watermarking technique in Internet Digital Traditional Music Museum. | N | N | Y | N | N | N | N | N |
| Chen et al. [4] | 2006 | Proposed E-Museum digital rights management. | N | Y | Y | N | Y | Y | Y | N |
| Chen [5] | 2008 | Proposed a mobile user to access enterprise digital resources. | N | N | Y | N | N | Y | Y | N |
| Chen [6] | 2010 | Supported a mobile user to access digital content. | N | Y | Y | N | N | Y | Y | N |
| Chen et al. [7] | 2014 | Authenticated users can use different mobile devices to access cloud services. | N | Y | Y | N | N | N | Y | N |
| Ma et al. [14] | 2018 | Proposed a blockchain-based, reliable, secure, efficient, and tamper-resistant digital content service. | Y | Y | Y | Y | Y | N | N | N |
| Ma et al. [15] | 2018 | Proposed two approaches for partially computing model (PCM) and non-computing of the model (NCM). | Y | N | N | Y | Y | N | N | N |
| Ma et al. [16] | 2020 | Proposed a blockchain-based decentralized trust management and secure usage control scheme of IoT big data. | Y | N | N | Y | Y | N | N | N |
| Hassan et al. [17] | 2020 | Proposed a DRM framework for protecting multimedia content. | N | Y | Y | N | N | N | N | N |
| Ours | 2020 | Proposed an authorization of museum's relic files. | Y | Y | Y | Y | Y | Y | Y | Y |

1: Blockchain-focused, 2: Comparative analysis with other approaches using tables, 3: Authentication, 4: Transparency, 5: Tamperproof, 6: Traceable, 7: Data format standardization, 8: Timeless, Y: Yes, and N: No.

*5.4. Limitations*

This mechanism can only operate normally based on the PKI (public key infrastructure) mechanism and the complete smart contract.

## 6. Conclusions

The digitalization of museums can avoid natural or man-made disasters and is also one of the sustainability trends of museums. The museum is a non-profit organization that shares intangible intellectual assets for the use of the whole of society. Its service boundaries will be spread through the Internet and become wider, expanding the museum's mission of social education sustainability; however, the digital archives of important cultural relics must still be protected through appropriate mechanisms. The main contributions of this work are as follows. This research is based on the mechanism of blockchain, smart contracts, and cryptography, and proposes an authorization protection model for museum digital rights.

The research results are based on the elliptic curve digital signature algorithm (ECDSA), blockchain, and smart contracts. The aim was to design a sustainable and traceable calculation method to achieve the museum's characteristics of public verification, transparency, non-tampering, traceability, non-repudiation, standardization of stored data, and timeliness of digital rights. In addition, the authorization of digital files can be fully used by the public or institutions in a reasonable range to perform the social education function of the museum at any time.

This study proposes that museum digitization and its security mechanisms have the following:

(1) Authorization of museum digitization—Museums control and manage the use of rights of digital content. Therefore, it is necessary to propose a new method for the preservation and authorization of museum digital content through blockchain and smart contracts, which can effectively display, store, and promote "important cultural relics and digital archives".

(2) The security mechanism of digital authorization—Museums must not only reflect social welfare, but the management of museum digital rights is a way to control and manage the use of digital content, through the authorization of cryptographic proxy re-encryption, to protect the digital content from abuse. The mechanism also ensures the security of digital assets.

(3) To increase the liveliness of the exhibition—Traditional exhibitions are usually physical exhibitions, while modern museums have diversified display modes. On the one hand, digital media technology can show museums as more vivid and with vivid interpretation of exhibition content. On the other hand, digital display methods can more easily break through various restrictions to make up for the lack of exhibition conditions. For example, the popular "Cloud Exhibition" is widely used in museums; it breaks through the limitations of time and space and is open to the public free of charge in a more vivid and three-dimensional form, which is a typical example of the effective use of digital assets.

(4) The transmission of social education—Museums are in the field of informal education, and the responsibility and mission of museums are to promote social education. Cultural relics in collections are the treasures of museums. Digitalization and establishment of digital archives can play an important role in the education resources of digital assets of museums.

(5) Development of cultural and creative commodities—Cultural and creative products are the in-depth manifestations of the value of cultural relics and can also be the derivatives of digital assets of museums. If the connotation of collectibles, cultural creativity, tourism, and other industries can be combined, the development of digital assets into creative goods is also one of the media of education communication.

**Author Contributions:** Conceptualization, Y.-C.W. and C.-L.C.; methodology, C.-L.C. and Y.-Y.D.; validation, Y.-C.W., C.-L.C., and Y.-Y.D.; investigation, Y.-C.W.; data analysis, C.-L.C.; writing—original draft preparation, Y.-Y.D.; writing—review and editing, Y.-C.W.; supervision, C.-L.C. All authors have read and agreed to the published version of the manuscript.

**Funding:** This research received no external funding.

**Institutional Review Board Statement:** This study only base on the theoretical basic research. It is not involving humans.

**Informed Consent Statement:** This study only base on the theoretical basic research. It is not involving humans.

**Data Availability Statement:** The data used to support the findings of this study are available from the corresponding author upon request.

**Conflicts of Interest:** The authors declare no conflict of interest.

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
