# Peer review of "Museum-Authorization of Digital Rights: A Sustainable and Traceable Cultural Relics Exhibition Mechanism"

_sustainability, doi:10.3390/su13042046_

Round 1
Reviewer 1 Report
This manuscript provides sufficient background and include relevant references. The research design is appropriate, and the methods adequately described. It is original and really interesting.
Author Response
Thanks for the reviewer’s positive comments.
Reviewer 2 Report
The paper discusses an approach to aplying elliptic curve digital signature algorithm (ECDSA), blockchain, and smart contracts to digital rights management in a cultural heritage context. The approach itself is interesting and responds to an well established need in the field of cultural heritage. However, the paper lacks in properly establishing the argument. The introduction is difficult to follow, discussing natural disasters and accidents and how they may cause destruction of physical artefacts, seemingly in an effort to build an argument for digitization. There is no need however to really build this argument. The need for digitization in cultural heritage is well establish and proven also by the millions of already existing digital assets. It would be more improtant for this paper to focus on the existing and not resolved need to properly manage the rights of such digitized content and build their argument on this. Discussion on this aspect could be expanded in the paper and founded on existing bibliography. Similarly, the conclusions make an argument for the benefits of digital elements and installations in physical exhibitions. Again, there is not need to establish these benefits, just maybe mention it briefly and point to relevant bibliography. The paper sturcture could also be improved. The introduction arguments are difficult to follow and possibly the comparison of existing approaches should be expanded in its own section, to also highlight the main contributions of this work, which are not clear. The language of the paper should be improved as in some cases the vocabulary and syntax makes it difficult to follow. (Lines 38 -39 "which as a form of disaster", for example). The purely algorithmic parts could probably be moved to an appendix.Author Response
- The paper discusses an approach to applying the elliptic curve digital signature algorithm (ECDSA), blockchain, and smart contracts to digital rights management in a cultural heritage context. The approach itself is interesting and responds to a well-established need in the field of cultural heritage. However, the paper lacks in properly establishing the argument. The introduction is difficult to follow, discussing natural disasters and accidents and how they may cause the destruction of physical artifacts, seemingly in an effort to build an argument for digitization. There is no need however to really build this argument. The need for digitization in cultural heritage is well established and proven also by the millions of already existing digital assets.
Authors’ response:
Thanks for the reviewer’s comments. As the reviewer’s argument, the cultural relics build digitization is an inevitable trend. This article aims to propose a sustainable and traceable exhibition mechanism for museums. One of the reasons is that the social education mission of museums can be achieved through the promotion of digital cultural relics. However, under this premise, digital cultural relics must be established before unpredictable natural or man-made disasters occur. Therefore, we will take the fire of Notre Dame Cathedral in Paris on April 15, 2019, as an example, which is a form of disaster. Therefore, whether the digitization of cultural relics is the main factor that will affect the operation of the museum itself and the sustainable promotion of the mission of social education. So, we put forward such a reminder in the introduction section as a description of the environmental background.
- It would be more important for this paper to focus on the existing and not resolved need to properly manage the rights of such digitized content and build their argument on this. Discussion on this aspect could be expanded in the paper and founded on the existing bibliography. Similarly, the conclusions make an argument for the benefits of digital elements and installations in physical exhibitions. Again, there is no need to establish these benefits, just maybe mention it briefly and point to relevant bibliography. The paper structure could also be improved. The introduction arguments are difficult to follow and possibly the comparison of existing approaches should be expanded in its section, to also highlight the main contributions of this work, which are not clear.
Authors’ response:
Thanks for the reviewer’s comments. To focus on the existing and not resolved need to properly manage the rights, we modify the part descriptions in the introduction section and use Table 1 to present a comparison of the existing digital right management surveys and detail the current research status in the introduction section. If these digitalized cultural relics exhibitions lack a complete authorization verification mechanism, these digital materials will be arbitrarily spread or even falsified. Based on this research foundation, we fix the demerits of past research to strengthen the analysis. In addition, this research also integrates blockchain technology to achieve more comprehensive and innovative applications. In the proposed research, we list the non-repudiation issue in Table 2, computation cost analysis in Table 3, and communication cost analysis in Table 4. Finally, we show the strength and quality of the work performed in Table 5. We follow the reviewer’s suggestion and cite the related works in our research goals on page 5. Through Table 1 to Table 5, the importance of digital rights management of the museum and the current insufficient research are provided step by step. This article introduces blockchain technology with innovative thinking to highlight the contribution of this work. We modified some content in page 2 with the red font.
- The language of the paper should be improved as in some cases the vocabulary and syntax make it difficult to follow. (Lines 38 -39 "which as a form of disaster", for example). The purely algorithmic parts could probably be moved to an appendix.
Authors’ response:
Thanks for the reviewer’s valuable suggestions. We have fixed the reviewer’s concerns and carefully check the typos through the full text. The algorithm is the procedure of the deduction in this article. After discussing with each other, the authors feel that it is more fluent to keep it in the original position. Therefore, to keep readable and related content, we suggest keeping the current state.

Reviewer 3 Report
- This review was not blind, as the authors' names are mentioned in the paper.
- The paper's overall theme is digital rights management in the field of ​​museums. The authors propose a scheme to improve and make sustainable the management, preservation and authorization of digital content in museums, in order to protect museum collections.
- I suggest that the title of the paper be changed to: Museum-Authorization of Digital Rights: A Sustainable and Traceable Cultural Relics Exhibition Mechanism
- The abstract and keywords are correct.
- The introduction contains the problem statement and a good literature review, some definitions of concepts and practical examples. This is correct. In line 84, insert before [14-17] "Some authors".
- Table 1 is fine. Good job.
- In line 133, a bibliographic reference should be inserted.
- Section 2 is also correct.
- Section 3 is fine. Figures 1 and 3 are also very good.
- The analysis and discussion are well done. Table 5 shows the strength and quality of the work performed.
- The conclusion is clear and objective.
- In general, the paper is of high quality.
Author Response
Reviewer 3 Comments:
- This review was not blind, as the authors' names are mentioned in the paper.
Authors’ response:
Thanks for the reviewer’s comments. In the future, we will pay attention to improving this missing.
- The paper's overall theme is digital rights management in the field of museums. The authors propose a scheme to improve and make sustainable the management, preservation, and authorization of digital content in museums, to protect museum collections.
Authors’ response:
Yes, it is a valuable issue.
- I suggest that the title of the paper be changed to Museum-Authorization of Digital Rights: A Sustainable and Traceable Cultural Relics Exhibition Mechanism
Authors’ response:
Thanks for the reviewer’s suggestion. We have changed the title to “Museum-Authorization of Digital Rights: A Sustainable and Traceable Cultural Relics Exhibition Mechanism”.
- The abstract and keywords are correct.
Authors’ response:
Thanks for the reviewer’s positive comments.
- The introduction contains the problem statement and a good literature review, some definitions of concepts, and practical examples. This is correct. In line 84, insert before [14-17] "Some authors".
Authors’ response:
We have modified it.
- Table 1 is fine. Good job.
Authors’ response:
Thanks for the reviewer’s positive comments.
- In line 133, a bibliographic reference should be inserted.
Authors’ response:
We have added the reference [18] in our manuscript.
- Section 2 is also correct.
Authors’ response:
- Section 3 is fine. Figures 1 and 3 are also very good.
Authors’ response:
Thanks for the reviewer’s positive comments.
- The analysis and discussion are well done. Table 5 shows the strength and quality of the work performed.
Authors’ response:
Thanks for the reviewer’s positive comments.
- The conclusion is clear and objective.
Authors’ response:
Thanks for the reviewer’s positive comments.
- In general, the paper is of high quality.
Authors’ response:
Thanks for the reviewer’s positive comments.

Reviewer 4 Report
Blockchain is a promising technology for establishing trust in digital systems, where network nodes do not necessarily trust each other. Cryptographic hash links and distributed consensus mechanisms ensure that the data stored on an immutable blockchain cannot be altered or deleted.
This paper presents a new method for the preservation and authorization of digital content in museums. The model proposed is based on the mechanism of blockchain, smart contracts, and 565 cryptography. According to the authors, the aim of the paper “was to design a sustainable and traceable calculation method to achieve 568 the museum's characteristics of public verification, transparency, non-tampering, traceability, non569 repudiation, standardization of stored data, and timeliness of digital rights”.
Strengths:
The paper presents an excellent organization and the points advanced in logical ways.
This paper provides an overview of the development of the area.
The presentation of the system architecture model provides an overview of the system.
The authors present the result of the model and a comparison of the proposed and existing digital right management surveys.
The documentation of sources and references are appropriate.
The tables and figures are useful and appropriate.
Weaknesses
The innovation and limitations of the study could be presented.
Proposed amendments
Table 1 is unformatted.
In my opinion, the codification of the model should have been presented in the Appendix.
Author Response
Reviewer 4 Comments:
Blockchain is a promising technology for establishing trust in digital systems, where network nodes do not necessarily trust each other. Cryptographic hash links and distributed consensus mechanisms ensure that the data stored on an immutable blockchain cannot be altered or deleted.
This paper presents a new method for the preservation and authorization of digital content in museums. The model proposed is based on the mechanism of blockchain, smart contracts, and 565 cryptography. According to the authors, the aim of the paper “was to design a sustainable and traceable calculation method to achieve 568 the museum's characteristics of public verification, transparency, non-tampering, traceability, non569 repudiation, standardization of stored data, and timeliness of digital rights”.
Strengths:
The paper presents an excellent organization and the points advanced in logical ways.
This paper provides an overview of the development of the area.
The presentation of the system architecture model provides an overview of the system.
The authors present the result of the model and a comparison of the proposed and existing digital right management surveys.
The documentation of sources and references are appropriate.
The tables and figures are useful and appropriate.
Authors’ response:
Many thanks for the reviewer's positive comments.
- Weaknesses
The innovation and limitations of the study could be presented.
Authors’ response:
Thanks for the reviewer’s comments.
- The innovation is described in Line 577-585 as follows.
“The main contributions of this work are as follows. This research is based on the mechanism of blockchain, smart contracts, and cryptography, and proposes an authorization protection model for museum digital rights. The research results are based on the elliptic curve digital signature algorithm (ECDSA), blockchain, and smart contracts. The aim was to design a sustainable and traceable calculation method to achieve the museum's characteristics of public verification, transparency, non-tampering, traceability, non-repudiation, standardization of stored data, and timeliness of digital rights. In addition, the authorization of digital files can be fully used by the public or institutions in a reasonable range to perform the social education function of the museum at any time.”
- The limitations are described in Line 568-570 (Section 5.4) as follows.
“5.4 Limitations
This mechanism can only operate normally based on the PKI (Public Key Infrastructure) mechanism and the complete smart contract.”
- Proposed amendments: Table 1 is unformatted.
Authors’ response:
We have reformatted Table 1.
- In my opinion, the codification of the model should have been presented in the Appendix.
Authors’ response:
Thanks for the reviewer’s comments. The algorithm is the procedure of the deduction in this article. After discussing with each other, the authors feel that it is more fluent to keep it in the original position. Therefore, to keep readable and related content, we suggest keeping the current state.
